# WaLRUS: Wavelets for Long-range Representation Using SSMs

**Hossein Babaei    Mel White    Sina Alemohammad    Richard G. Baraniuk**
Department of Electrical and Computer Engineering, Rice University
{hb26,mel.white,sa86,richb}@rice.edu

## Abstract

State-Space Models (SSMs) have proven to be powerful tools for modeling long-range dependencies in sequential data. While the recent method known as HiPPO has demonstrated strong performance, and formed the basis for machine learning models S4 and Mamba, it remains limited by its reliance on closed-form solutions for a few specific, well-behaved bases. The SaFARi framework generalized this approach, enabling the construction of SSMs from arbitrary frames, including non-orthogonal and redundant ones, thus allowing an infinite diversity of possible "species" within the SSM family. In this paper, we introduce WaLRUS (Wavelets for Long-range Representation Using SSMs).We compare WaLRUS to HiPPO-based models, and demonstrate improved accuracy and more efficient implementations for online function approximation tasks.

## 1 Introduction

Sequential data is foundational to many machine learning tasks, including natural language processing, speech recognition, and video understanding [1–3]. These applications require models that can effectively process and retain information over long time horizons. A central challenge in this setting is the efficient representation of long-range dependencies in a way that preserves essential features of the input signal for downstream tasks, while remaining computationally tractable during both training and inference [4].

Recurrent neural networks (RNNs) are traditional choices for modeling sequential data, but struggle with long-term dependencies due to vanishing or exploding gradients during backpropagation through time [4–6]. While gated variants like LSTMs [7] and GRUs [8] mitigate some issues, they require significant tuning and lack compatibility with parallel processing, hindering scalability.

State-space models (SSMs) offer a linear and principled framework for encoding temporal information, and have re-emerged as a powerful alternative for online representation of sequential data [9–16]. By design, they enable the online computation of compressive representations that summarize the entire input history using a fixed-size state vector, ensuring a constant memory footprint regardless of sequence length. A major breakthrough came with HiPPO (High-order Polynomial Projection Operators), which reformulates online representation as a function approximation problem using orthogonal polynomial bases [9]. This approach underpins state-of-the-art models like S4 and Mamba, enabling compact representations for long-range dependencies [10, 11].

However, existing SSMs primarily rely on Legendre and Fourier bases, which, although effective for smooth or periodic signals, struggle with non-stationary and localized features [9, 10]. These challenges are especially evident in domains such as audio, geophysics, and biomedical signal processing, where rapid transitions and sparse structure are common.

To address this limitation, the SaFARi framework (State-Space Models for Frame-Agnostic Representation) extends HiPPO to arbitrary frames, including non-orthogonal and redundant bases [13, 14, 17].

39th Conference on Neural Information Processing Systems (NeurIPS 2025).

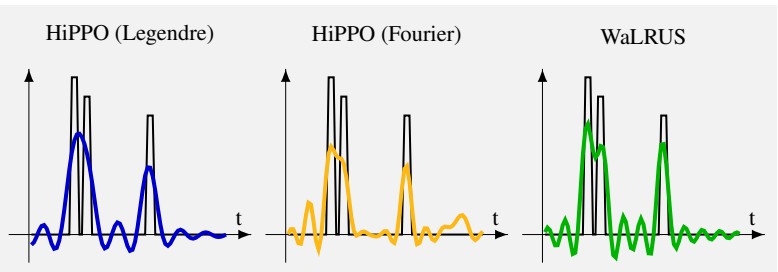

Figure 1: An input signal comprising three random spikes is sequentially processed by SSMs and reconstructed after observing the entire input. Only the wavelet-based SSM constructed using WaLRUS can clearly distinguish adjacent spikes.

This generalization enables SSM construction from any frame via numerical solutions of first-order linear differential equations, preserving HiPPO's memory efficiency and update capabilities without closed-form restrictions.

In this paper, we leverage the SaFARi method with wavelet frames to introduce a new model, WaLRUS (Wavelets for Long-range Representation Using SSMs). We derive our model using Daubechies wavelets with two variants: scaled-WaLRUS and translated-WaLRUS, designed for capturing non-smooth and localized features through compactly supported, multi-resolution wavelet decompositions [18]. These properties allow WaLRUS to retain fine-grained signal details typically lost by polynomial-based models.

We also provide a comparative analysis of WaLRUS and existing HiPPO variants (see Fig. 1). Empirical results demonstrate that the wavelet-based WaLRUS model consistently outperforms Legendre and Fourier-based HiPPO models in reconstruction accuracy, especially on signals with sharp transients. Furthermore, WaLRUS has been experimentally observed to be stably diagonalizable, which is the key enabler of efficient convolution-based implementations and parallel computation [13, 14].

These results highlight the practical advantages of WaLRUS models, particularly in scenarios where signal structure varies across time and scale. By bridging multiscale signal analysis and online function approximation, WaLRUS opens new directions for modeling complex temporal phenomena across disciplines.

## 2   Background

Recent advances in machine learning, computer vision, and large language models have pushed the frontier of learning from long sequences of data. These applications demand models that can (1) generate compact representations of input streams, (2) preserve long-range dependencies, and (3) support efficient online updates.

Classical linear methods, such as the Fourier transform, offer compact representations in the frequency domain [19–23]. However, they are ill-suited for online processing: each new input requires recomputing the entire representation, making them inefficient for streaming data and limited in their memory horizon. Nonlinear models like recurrent neural networks (RNNs) and their gated variants (LSTMs, GRUs) have been more successful in sequence modeling, but they face well-known issues such as vanishing/exploding gradients and limited parallelization [4–6, 8]. Moreover, their representations are task-specific, and not easily repurposed across different settings.

To resolve these issues, the HiPPO framework [9] casts online function approximation as a continuous projection of the input $u(t)$ onto a linear combination of the given basis functions $\mathcal{G}$. At every time $T$, it produces a compressed state vector $\vec{c}(T)$ that satisfies the update rule:

$$\frac{d}{dT}\vec{c}(T) = -A_{(T)}\vec{c}(T) + B_{(T)}u(T). \tag{1}$$

Here, $A_{(T)}$ and $B_{(T)}$ are derived based on the choice of polynomial basis and measure $\mu(t)$, which defines how recent history is weighted. Two commonly used measures are:

$$\mu_{tr}(t) = \frac{1}{\theta} \mathbb{1}_{t \in [T-\theta, T]}, \quad \mu_{sc}(t) = \frac{1}{T} \mathbb{1}_{t \in [0, T]}. \tag{2}$$

The translated measure $\mu_{tr}$ emphasizes recent history within a sliding window of length $\theta$, while the scaled measure $\mu_{sc}$ compresses the entire input history into a fixed-length representation.

Despite its strengths, HiPPO is restricted to only a few bases (e.g., Legendre, Fourier), and deriving $A(t)$ and $B(t)$ in closed form is only tractable for specific basis-measure combinations.

SaFARi addressed this limitation by generalizing online function approximation to any arbitrary frame [17]. A frame $\Phi(t)$ is a set of elements $\{\phi_i(t)\}$ such that one can reconstruct any input $g(t)$ by knowing the inner products $\langle g(t), \phi_i(t) \rangle$. For a given frame $\Phi$, its complex conjugate $\overline{\Phi}$, and its dual $\widetilde{\Phi}$, the scaled-SaFARi produces an SSM with $A$ and $B$ given by:

$$\frac{\partial}{\partial T} \vec{c}(T) = -\frac{1}{T} A \vec{c}(T) + \frac{1}{T} B u(T), \quad A_{i,j} = \delta_{i,j} + \int_0^1 t' \frac{\partial}{\partial t} \overline{\phi}_i \bigg|_{t=t'} \widetilde{\phi}_j(t') dt', \quad B_i = \overline{\phi}_i(1) \tag{3}$$

while the translated-SaFARi produces an SSM with the $A$ and $B$ given by:

$$\frac{\partial}{\partial T} \vec{c}(T) = -\frac{1}{\theta} A \vec{c}(T) + \frac{1}{\theta} B u(T), \quad A_{i,j} = \overline{\phi}_i(0) \widetilde{\phi}_j(0) + \int_0^1 \frac{\partial}{\partial t} \overline{\phi}_i \bigg|_{t=t'} \widetilde{\phi}_j(t') dt', \; B_i = \overline{\phi}_i(1) \tag{4}$$

In the appendix, we provide a some theoretical background on Eq. 3 and Eq. 4 from [17].

**Incremental update of SSMs**: The differential equation in Eq. 1 can be solved incrementally. Following [9], we adopt the Generalized Bilinear Transform (GBT) [24] given by Eq. 5 for its superior numerical accuracy in first order SSMs.

$$c(t + \Delta t) = (I + \delta t \alpha A_{t+\delta t})^{-1} \left[ (I - \delta t(1-\alpha) A_t) c(t) + \delta t B(t) u(t) \right] \tag{5}$$

**Diagonalization of $A$**: Each GBT step involves matrix inversion and multiplication. If $A(t)$ has time-independent eigenvectors (e.g., $A(t) = g(t)A$), it can be diagonalized as $A(t) = V \Lambda(t) V^{-1}$, allowing a change of variables $\widetilde{c} = V^{-1} c$ and $\widetilde{B} = V^{-1} B(t)$, yielding:

$$\frac{\partial}{\partial t} \widetilde{c} = -\Lambda(t) \widetilde{c} + \widetilde{B} u(t), \tag{6}$$

This reduces each update to elementwise operations, significantly lowering computational cost.

## 2.1 Wavelet Frames

Wavelet frames offer a multiresolution analysis that captures both temporal and frequency characteristics of signals, making them particularly effective for representing non-stationary or long-range dependent data [25]. Initiated by [26] and formalized by [27], wavelet theory gained prominence with Ingrid Daubechies' seminal work [28], which introduced compactly supported orthogonal wavelets. Since then, wavelets have played a central role in modern signal processing [29].

Wavelet analysis decomposes a signal $f(t)$ into dilations and translations of a mother wavelet $\psi(t)$, enabling simultaneous localization in time and frequency. The *discrete wavelet transform* is

$$W(j, k) = \int_{-\infty}^{\infty} f(t) \psi_{j,k}^*(t) \, dt, \quad \psi_{j,k}(t) = \frac{1}{\sqrt{2^{-j}}} \psi\left(\frac{t-k}{2^{-j}}\right).$$

Unlike global bases such as Fourier or polynomials, which struggle with localized discontinuities, wavelets provide sparse representations of signals with singularities, such as jumps or spikes [18, 30]. Their local support yields small coefficients in smooth regions and large coefficients near singularities, enabling efficient compression and accurate reconstruction. These properties make wavelet frames a natural and powerful choice for time-frequency analysis in a wide range of practical applications.

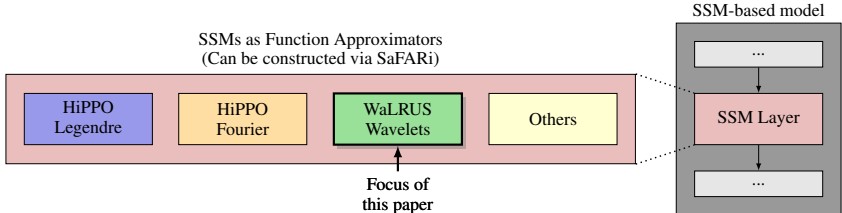

Figure 2: A diagram of the relationships between HiPPO, SaFARi, WaLRUS (this work), and SSM-based models such as S4 and Mamba. The focus of this work is on the development of a wavelet-based SSM in a function approximation task, which could later be used as a drop-in replacement for the SSM layer in a learned model.

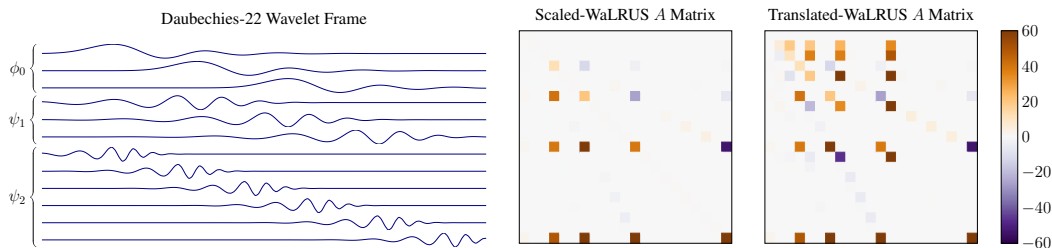

Figure 3: Left: Elements of a Daubechies-22 wavelet frame, with father wavelet $\phi$, mother wavelet $\psi$, and two scales. Right: The scaled and translated $A$ matrices for WaLRUS with $N = 21$.

## 3 WaLRUS: Wavelet-based SSMs

Daubechies wavelets [18, 28] provide a particularly useful implementation of a SaFARi SSM. While there are different types of commonly used wavelets, Daubechies wavelets are of particular interest in signal representation due to their maximal vanishing moments over compact support.

To construct the frame, we use the usual dyadic scaling for multiresolution analysis; that is, scaling the mother wavelets by a factor of two at each level. For each scale, different shifts along the x-axis are introduced. Compressive wavelet frames are truncated versions of wavelet frames that contain only a few of the coarser scales, and introduce overlapping shifts to keep the expressivity and satisfy the frame condition (See Mallat, [29]). The interplay between the retained scales and the minimum required overlap to maintain the expressivity is extensively studied in the wavelet literature [18, 28, 29]. If there is excess overlap in shifts, the wavelet frame becomes redundant, and redundancy has advantages in expressivity and robustness to noise.

Figure 3, left, gives a visual representation of how we construct such a frame. The frame consists of shifted copies of the father wavelet $\phi$ at one scale, and shifted copies of a mother wavelet $\psi$ at different scales, with overlaps that introduce redundancy. Figure. 3, right, shows the resulting $A$ matrices for the scaled and translated WaLRUS.[1]

Some recent works [31, 32] has conceptually connected the use of wavelets and SSM-based models (namely Mamba). These efforts are fundamentally distinct from ours in that they perform a multi-resolution analysis on the input to the model only. No change is made to the standard Mamba SSM layer.

This work, on the other hand, is the first to challenge the ubiquity of the Legendre-based SSM, and present alternative wavelet-based machinery for the core of powerful models like Mamba. WaLRUS could be used as a drop-in replacement for any existing SSM-based framework. However, before simply substituting a part in a larger system, we must first justify how and why a different SSM can improve performance. This paper presents a tool that stands alone as an online function approximator, and also provides a foundational building block for future integration in SSM-based models.

---

[1]Code to generate the matrices is available at the following repository: `https://github.com/echbaba/walrus`.

## 3.1 Redundancy of the wavelet frame and size of the SSM

In contrast to orthonormal bases, redundant frames allow more than one way to represent the same signal. This redundancy arises from the non-trivial null space of the associated frame operator, meaning that multiple coefficient vectors can yield the same reconstructed function. Although the representation is not unique, it is still perfectly valid, and this flexibility offers several key advantages in signal processing. In particular, redundancy can improve robustness to noise, enable better sparsity for certain signal classes, and enhance numerical stability in inverse problems [33–35].

We distinguish between the total number of frame elements $N_{\text{full}}$ and the effective dimensionality $N_{\text{eff}}$ of the subspace where the meaningful representations reside. In other words, while the frame may consist of $N_{\text{full}}$ vectors, the actual information content lies in a lower-dimensional subspace of size $N_{\text{eff}}$. This effective dimensionality can be quantified by analyzing the singular-value spectrum of the frame operator [29, 33].

For the WaLRUS SSMs described in this work, we first derive $A_{N_{\text{full}}}$ using all elements of the redundant frame. We then diagonalize $A$ and reduce it to a size of $N_{\text{eff}}$. This ensures that different frame choices, whether orthonormal or redundant, can be fairly and meaningfully compared in terms of computational cost, memory usage, and approximation accuracy. The exact relationship between the wavelet frame and the resulting $N_{\text{eff}}$ of the $A$ matrix depends not only on the overlap of the shifts in the frame, but also on the type (and order) of chosen wavelet, and number of scales. Determining the "optimal" overlap or $N_{\text{eff}}$ is application-specific and an area for future research.

## 3.2 Computational complexity of WaLRUS

For a sequence of length $L$, scaled-SaFARi has $O(N^3 L)$ complexity due to solving an $N$-dimensional linear system at each step, while translated-SaFARi can reuse matrix inverses, and thus has $O(N^2 L)$ complexity, assuming no diagonalization [17]. When the state matrix $A$ is diagonalizable, the complexity reduces to $O(NL)$ and can further accelerate to $O(L)$ with parallel processing on independent scalar SSMs.

We observe that each of the scaled and translated WaLRUS SSMs we implemented, regardless of dimension, were stably diagonalizable. Further research is required to determine whether Daubechies wavelets will always yield diagonalizable SSMs. Legendre-based SSMs, on the other hand, are not stably diagonalizable [9]. Although [9] proposed a fast sequential HiPPO-LegS update to achieve $O(NL)$ complexity, [17] showed that it cannot be parallelized to $O(L)$. Moreover, no efficient sequential update exists for HiPPO-LegT, leaving Legendre-based SSMs at a disadvantage during inference when sequential updates are needed.

As sequence length increases, step-wise updates become a bottleneck, especially during training when the entire sequence is available upfront. This can be mitigated by using convolution kernels instead of sequential updates. Precomputing the convolution kernel and applying it via convolution accelerates computation, leveraging GPU-based parallelism to achieve $O(\log L)$ run-time complexity for diagonalizable SSMs. This optimization is feasible for both WaLRUS and Fourier-based SSMs. Although Legendre-based SSMs can attain similar asymptotic complexity through structured algorithms [10, 12], their nondiagonal nature prevents decoupling into $N$ independent SSMs.

## 3.3 Representation errors in the translated WaLRUS

Truncated representations in SSMs inevitably introduce errors, as discarding higher-order components limits reconstruction fidelity [17]. SaFARi only investigated these errors for scaled SSMs, leaving their approximation accuracy unquantified. Visualizing the convolution kernels generated by different SSMs offers some insight into the varying performance of different SSMs on the function approximation task. An "ideal" kernel would include a faithful representation for each element of the basis or frame from $T = 0$ to $T = W$, where $W$ is the window width, and it would contain no non-zero elements between $W$ and $L$. However, certain bases generate kernels with warping issues, as illustrated in Fig. 4.

The HiPPO-LegT kernel loses coefficients due to warping within the desired translating window (see areas B and C of Fig. 4). For higher degrees of Legendre polynomials, the kernel exhibits an all-zero region at the beginning and end of the sliding window. This implies that high-frequency information in the input is not captured at the start or end of the sliding window, and the extent of this dead zone

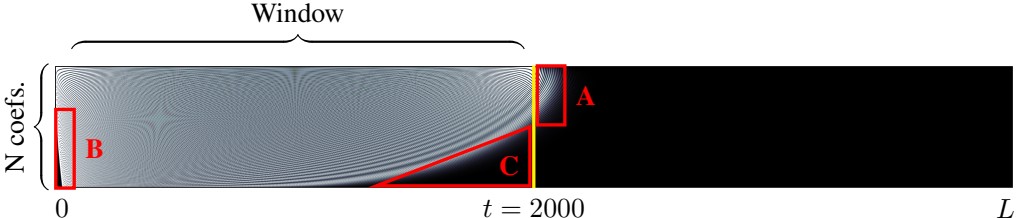

Figure 4: The kernel generated by HiPPO-LegT with window size $W = 2000$ and representation size $N = 500$. Three key non-ideal aspects of the kernel are noticeable. **A)** poor localization due to substantial non-zero values outside $W$, **B)** coefficient loss from at bottom left of the kernel, and **C)** coefficient loss at the bottom right of the kernel for $t \in (1500, 2000)$.

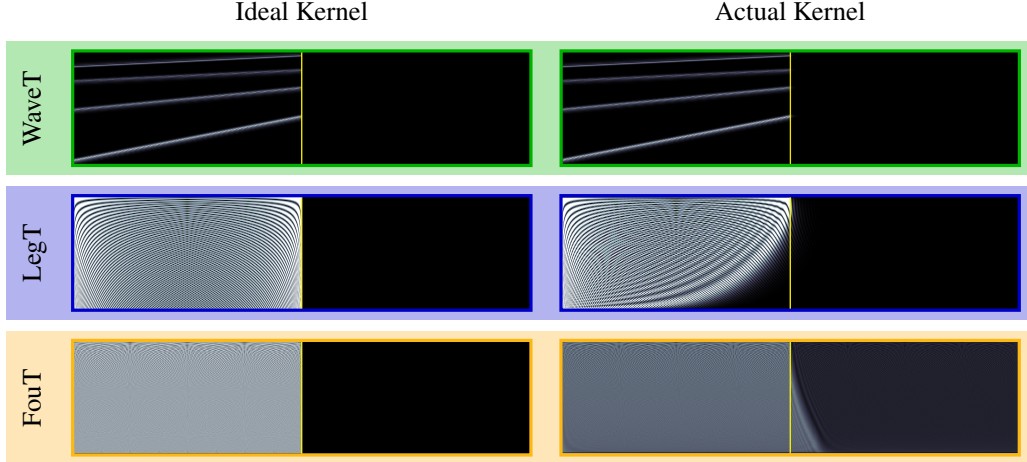

Figure 5: **Left:** The ideal kernels, which yield zero representation error, are shown for Translated-WaLRUS (using the D22 wavelet), HiPPO-LegT, and HiPPO-FouT. **Right:** The corresponding kernels generated by the translated models are presented for comparison. WaveT has superior localization within the window of interest compared to HiPPO-LegT and HiPPO-FouT.

increases with higher frequencies. The translated Fourier kernel primarily suffers from the opposite problem: substantial nonzero elements outside the kernel window indicate that LegT struggles to effectively "forget" historical input values. Thus contributions from input signals outside the sliding window appear as representation errors. LegT also has this problem, to a lesser extent–see area A of Fig. 4 for a closer view of the kernel.

A visual inspection of Fig. 5 reveals that the translated-WaLRUS kernel closely matches the idealized version, whereas both FouT and LegT exhibit significant errors in their computed kernels. We emphasize that the issues observed with LegT and FouT arise from inherent limitations of the underlying SSMs themselves and are not due to the choice of input signal classes.

## 4 Experiments

The following section deploys the WaLRUS SSM on synthetic and real signals for the task of function approximation, comparing its performance with extant models in the literature. We will evaluate performance in MSE as well as their ability to track important signal features like singularities, and show that using WaLRUS can have an edge over the state-of-the-art polynomial-based SSMs.

To benchmark WaLRUS against state-of-the-art SSMs, we implement two variants: *Scaled-WaLRUS* and *Translated-WaLRUS*, which we will call WaveS and WaveT respectively, following HiPPO's convention. These models are compared against the top-performing HiPPO-based SSMs. Further details on the wavelet frames used in each experiment are provided in Appendix A.2.4, and code can be found at `https://github.com/echbaba/walrus`.

We conduct experiments on the following datasets:

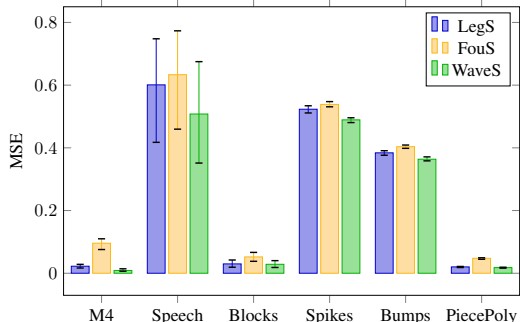

| Dataset | LegS | FouS | WaveS |
|---------|------|------|-------|
| M4 | 0% | 0.47% | **99.53%** |
| Speech | 4.25% | 0% | **95.75%** |
| Blocks | 0% | 0% | **100%** |
| Spikes | 0% | 0% | **100%** |
| Bumps | 0% | 0% | **100%** |
| Piecepoly | 1.00% | 0% | **99.00%** |

Figure 6: Comparing reconstruction MSE between WaveS, LegS, and FouS. Error bars represent the first and third quantile of MSE. WaveS produces the lowest MSE in each dataset.

Table 1: Percent of tests where each basis had the lowest overall MSE.

**M4 Forecasting Competition [36]:** A diverse collection of univariate time series with varying sampling frequencies taken from domains such as demographic, finance, industry, macro, micro, etc.

**Speech Commands [37]:** A dataset of one-second audio clips featuring spoken English words from a small vocabulary, designed for benchmarking lightweight audio recognition models.

**Wavelet Benchmark Collection [38]:** A synthetic benchmark featuring signals with distinct singularity structures, such as Bumps, Blocks, Spikes, and Piecewise Polynomials. We generate randomized examples from each class, with further details and visualizations provided in Appendix A.2.2.

## 4.1 Comparisons among frames

We note that no frame is universally optimal for all input classes, as different classes of input signals exhibit varying decay rates in representation error. However, due to the superior localization and near-optimal error decay rate of wavelet frames, wavelet-based SSMs consistently show an advantage over Legendre and Fourier-based SSMs across a range of real-world and synthetic signals. These experiments position WaLRUS as a powerful and adaptable approach for scalable, high-fidelity signal representation.

### 4.1.1 Experimental setup

The performance of SSMs in online function approximation can be evaluated several ways. One metric is the mean squared error (MSE) of the reconstructed signal compared to the original. In the following sections, we compare the overall MSE for SSMs with a scaled measure, and the running MSE for SSMs with a translated measure.

Additionally, in some applications, the ability to capture *specific features* of a signal may be of greater interest than the overall MSE. As an extreme case, consider a signal that is nearly always zero, but contains a few isolated spikes. If our estimated signal is all zero, then the MSE will be small, but all of the information of interest has been lost.

In all the experiments, we use equal SSM sizes $N_{\text{eff}}$, as described in Sec. 3.1.

### 4.1.2 Function approximation with the scaled measure

In this experiment, we construct Scaled-WaLRUS, HiPPO-LegS, and HiPPO-FouS with equal effective sizes (see Appendix A.2.4). Frame sizes are empirically selected to balance computational cost and approximation error across datasets.

Fig. 6 shows the average MSE across random instances of multiple datasets. Not only is the average MSE lowest for WaLRUS for all datasets, but even where there is high variance in the MSE, all methods tend to keep the same *relative* performance. That is, the overlap in the error bars in Fig. 6 does not imply that the methods are indistinguishable; rather, for a given instance of a dataset, the MSE across all three SSM types tends to shift together, maintaining the MSE ordering WaveS <

| Dataset: | | Spikes | | | Bumps | | |
|---|---|---|---|---|---|---|---|
| Basis/Frame: | | Legendre | Fourier | Wavelets | Legendre | Fourier | Wavelets |
| Scaled | Peaks missed | 2.5% | 0.62% | **0%** | 0.29% | 0.30% | **0%** |
| | False peaks | 1.6% | 1.6% | **0.01%** | 0.3% | 1.9% | **0%** |
| | Instance-wise wins | 76% | 92.9% | **100%** | 97.1% | 96.9% | **100%** |
| | Relative amplitude error | 16.2% | 11.8% | **5.5%** | 12.4% | 16.2% | **6.5%** |
| | Average displacement | 18.8 | 32.0 | **10.0** | 12.7 | 33.7 | **7.1** |
| Translated | Peaks missed | 6.4% | 13.0% | **0.27%** | 1.12% | 29.76% | **0.08%** |
| | False peaks | 1.1% | **0.05%** | 0.22% | 0.43% | 0.28% | **0.20%** |
| | Instance-wise wins | 36.9% | 13.65% | **99.95%** | 85.1% | 0.2% | **100%** |
| | Relative amplitude error | 19.6% | 28.4% | **3.5%** | 6.9% | 28.4% | **2.5%** |
| | Average displacement | 6.0 | 5.4 | **4.3** | 5.5 | 5.8 | **4.8** |

Table 2: Performance comparison of WaLRUS-Wavelets, HiPPO-Legendre, and HiPPO-Fourier for peak detection with the translated measure. WaLRUS shows a significant advantage in successfully remembering singularities over HiPPO SSMs.

LegS < FouS. To highlight this result, the percentage of instances where each SSM had the best performance is also provided in Table 1.

The representative power of WaLRUS is attributed to its ability to minimize truncation and mixing errors by selecting frames that capture signal characteristics with higher fidelity. See [17] for further details.

### 4.1.3 Peak detection with the scaled measure

In this experiment, we aim to detect the locations of random spikes in input sequences using Scaled-WaLRUS, FouS, and LegS, all constructed with an equal sizes. We generate random spike sequences, add Gaussian noise (SNR = 0.001), and compute their representations with Daubechies wavelets, Legendre polynomials, and Fourier series. The reconstructed signals are transformed into wavelet coefficients, and spike locations are identified following the method in [30].

To evaluate performance, we compare the relative amplitude and displacement of detected spikes with their ground truth (see Fig.7). This process is repeated for 1000 random sequences, each containing 10 spikes. Table 2 summarizes the average number of undetected spikes for each SSM and the instance-wise win percentage, representing the number of instances where each SSM had fewer or equal misses peaks than the other SSMs. Note that these percentages do not sum to 100, as some instances result in identical spike detection across all models.

As shown in Table 2, WaveS misses significantly fewer spikes than FouS and LegS, with lower displacement errors and reduced amplitude loss. Figure 1 illustrates an example where WaLRUS successfully captures closely spaced spikes that are missed by LegS and FouS, demonstrating its superior time resolution.

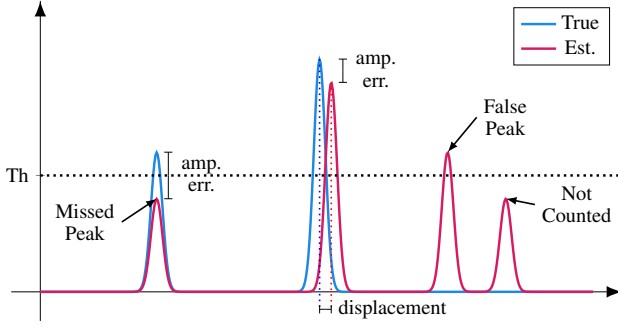

Figure 7: Illustration of the metrics to evaluate performance of SSMs on different datasets in Table 2.

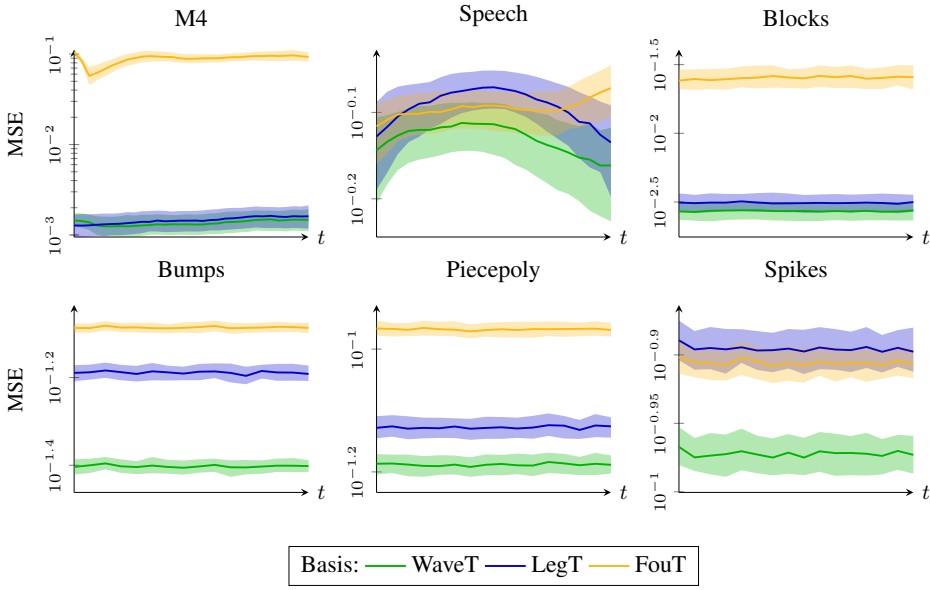

Figure 8: For each dataset, the median and $(0.4, 0.6)$ quantile of running reconstruction MSE across different instances is demonstrated in different colors for WaveT, LegT, and FouT. WaveT captures information in the input signals with a higher fidelity than LegT and FouT.

#### 4.1.4 Function approximation with the translated measure

In this experiment, we construct WaveT, LegT, and FouT SSMs, all with equal effective sizes (see Appendix A.2.4). The chosen effective sizes are smaller than those we used for the scaled measure since the translated window contains lower frequency content within each window, making it possible to reconstruct the signal with smaller frames. Then, for each instance of input signal, the reconstruction MSE at each time step is calculated and plotted in Fig. 8.

For each input signal instance, we compute the running MSE at each time step, as shown in Fig. 8. This plot represents how the MSE evolves over time across multiple instances, providing a comparison of running MSEs for each SSM. The results demonstrate that Translated-WaLRUS consistently achieves slightly better fidelity than LegT and significantly outperforms FouT across all datasets.

As discussed in Section 3.3, the reconstruction error stems from two main factors: (1) non-idealities in the translated SSM kernel, affecting its ability to retain relevant information within the window while effectively forgetting data outside it (see Fig. 4), and (2) the extent to which these fundamental non-idealities are activated by the input signal. For example, signals with large regions of zero values are less impacted by kernel inaccuracies, as the weights outside the kernel contribute minimally to reconstruction.

WaveT achieves a modest, and in some cases negligible MSE improvement over LegT (e.g., M4 and Blocks). However, the kernel-based limitations highlighted in Section 3.3 may have a more pronounced effect on longer sequences or different datasets.

#### 4.1.5 Peak detection with the translated measure

In this experiment, we evaluate the ability of WaveT, FouT, and LegT to retain information about singularities in signals, following the setup in Section 4.1.3, but with a translated SSM. We generate 2,000 random sequences, each containing 20 spikes. The average number of undetected spikes for each SSM, along with instance-wise win percentages, is reported in Table 2. As in the scaled measure experiment, the percentages do not sum to 100 due to ties across SSMs. Table 2 shows that WaveT consistently outperforms FouT and LegT, with fewer missed peaks, reduced displacement, and less amplitude loss.

# 5   Limitations

In this work we have implemented only one type of wavelet (Daubechies-22), as our purpose is to introduce practical and theoretical reasons to replace polynomial SSMs with wavelet SSMs. Other wavelets (biorthogonal, coiflets, Morlets, etc.) could also be used, with some caveats. First, we require a differentiable frame [17], so nondifferentiable wavelets like Haar wavelets or other lower-order Daubechies and Coiflets cannot be used with this method. Second, the redundancy of the frame (and the resulting $N_{\text{eff}}$ of the $A$ matrix) depends on the shape of the wavelet's function and the chosen shifts and scales of this function. Other wavelet types, and other choices of shift and scale, may exhibit better or worse performance and dimensionality reduction, and this is an important question for future work.

Additionally, we emphasize that the choice of frame is application-dependent. If the signal is known to be smooth and periodic, a wavelet-based SSM is not likely to outperform a Fourier-based SSM, for example. The introduction of WaLRUS is not intended to be a one-size-fits-all model, but rather a broadly-applicable tool that combines compressive online function-approximation SSMs with the expressive power of wavelets.

# 6   Conclusions

We have demonstrated in this paper how function approximation with SSMs, initially proposed by [9] and subsequently extended to general frames, can be improved using wavelet-based SSMs. SSMs constructed with wavelet frames can provide higher fidelity in signal reconstruction than the state-of-the-art Legendre and Fourier-based SSMs over both scaled and translated measures. Future work will explore alternate wavelet families, and the trade-offs in effective size, frequency space coverage, and representation capabilities of different frames.

Moreover, since the Legendre-based HiPPO SSM forms the core of S4 and Mamba, and WaLRUS provides a drop-in replacement for HiPPO, WaLRUS could be used to initialize SSM-based machine learning models–potentially providing more efficient training. As AI becomes ubiquitous, and the demand for computation explodes, smarter and more task-tailored ML architectures can help mitigate the strain on energy and environmental resources.

## Acknowledgments

Special thanks to T. Mitchell Roddenberry for fruitful conversations and insights. This work was supported by NSF grants CCF-1911094, IIS-1838177, and IIS-1730574; ONR grants N00014-18-12571, N00014-20-1-2534, N00014-18-1-2047, and MURI N00014-20-1-2787; AFOSR grant FA9550-22-1-0060; DOE grant DE-SC0020345; and ONR grant N00014-18-1-2047. Additional support was provided by a Vannevar Bush Faculty Fellowship, the Rice Academy of Fellows, and Rice University and Houston Methodist 2024 Seed Grant Program.

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

# A   Appendix

## A.1   SaFARi derivation for arbitrary frame

Where HiPPO [9] provided closed-form solutions to construct $A$ and $B$ for a few polynomial bases, SaFARi [17] introduced a method to build $A$ and $B$ from any arbitrary frame. The derivations provided below follow [17], and are given here as convenient reference for the reader.

Take a signal $f$ and frame $\psi$. To get a vector of weights representing a signal on a basis, we use the inner product:

$$c_n = \int f(t)\overline{\phi(t)}dt \tag{A.1}$$

So at some time $T$, we scale the magnitude of $f(t)$ and stretch the basis to match the length of $f$:

$$c_n(T) = \int_{t_0}^{T} f(t)\left(\frac{1}{T-t_0}\right)\overline{\phi\left(\frac{t-t_0}{T-t_0}\right)}dt \tag{A.2}$$

We are actually interested in the **change** in c. We will take the partial derivative with respect to $T$, since the coefficients update at each new time $T$. Call the start time $t_0$: this is 0 for the scaling case, and $t_0$ varies with the windowed case. If we call the size of the window $\theta$, then $t_0 = T - \theta$. The derivation below will be a generic version, then we will separate the two cases.

$$\frac{d}{dT}c_n(T) = \frac{d}{dT}\int_{t_0}^{T} f(t)\left(\frac{1}{T-t_0}\right)\overline{\phi\left(\frac{t-t_0}{T-t_0}\right)}dt \tag{A.3}$$

We note that this is the partial derivative of an integral bounded by two variables. Thus we call on Leibniz' integration rule and find:

$$\frac{d}{dT}c_n(T) = f(T)\left(\frac{1}{T-t_0}\right)\overline{\phi(1)}\frac{\delta}{\delta T}(T) - f(t_0)\left(\frac{1}{T-t_0}\right)\overline{\phi(0)}\frac{\delta}{\delta T}(t_0)$$
$$+ \int_{t_0}^{T} f(t)\underbrace{\frac{\delta}{\delta T}\left[\left(\frac{1}{T-t_0}\right)\overline{\phi\left(\frac{t-t_0}{T-t_0}\right)}\right]}_{\overline{h(t)}}dt \quad \text{(A.4)}$$

Some manipulation of the $h(t)$ term yields:

$$h(t) = \left(\frac{1}{T-t_0}\right)\left[-\frac{\delta(t_0)}{\delta T}\left(\frac{1}{T-t_0}\right)\phi'\left(\frac{t-t_0}{T-t_0}\right) - \left(1-\frac{\delta(t_0)}{\delta T}\right)\left(\frac{t-t_0}{T-t_0}\right)\phi'\left(\frac{t-t_0}{T-t_0}\right)\right]$$
$$- \left(\frac{1}{T-t_0}\right)\left[\left(\frac{1-\frac{\delta(t_0)}{\delta T}}{T-t_0}\right)\phi\left(\frac{t-t_0}{T-t_0}\right)\right] \tag{A.5}$$

Our h(t) term now has the derivative of our basis ($\phi'$) in it, but we'd like to be able to combine combine terms with $\phi$. Therefore we can make a mapping from $\phi' \to \phi$ using the dual, $\widetilde{\phi}$:

$$\phi'\left(\frac{t-t_0}{T-t_0}\right) = \underbrace{\left\langle \phi'\left(\frac{t-t_0}{T-t_0}\right), \widetilde{\phi}\left(\frac{t-t_0}{T-t_0}\right)\right\rangle}_{P}\phi\left(\frac{t-t_0}{T-t_0}\right) \tag{A.6}$$

Likewise:

$$(t-t_0)\phi'\left(\frac{t-t_0}{T-t_0}\right) = \underbrace{\left\langle (t-t_0)\,\phi'\left(\frac{t-t_0}{T-t_0}\right), \widetilde{\phi}\left(\frac{t-t_0}{T-t_0}\right)\right\rangle}_{P_t}\phi\left(\frac{t-t_0}{T-t_0}\right) \tag{A.7}$$

This lets us do another simplification of $h(t)$, and group all the functions of $\phi$. Let's also call $T - t_0 = \theta$ to save some space.

$$h(t) = \frac{1}{\theta}\phi\left(\frac{t - t_0}{T - t_0}\right)\left[-\frac{\delta(t_0)}{\delta T}\frac{1}{\theta}P - \left(1 - \frac{\delta(t_0)}{\delta T}\right)\frac{1}{\theta}P_t - \frac{1}{\theta}\left(1 - \frac{\delta(t_0)}{\delta T}\right)\right] \tag{A.8}$$

Now we can return to Eq. A.4. P is not a function of $t$, so it can be moved outside the integral. For the measures we are looking at, $\frac{d}{dT}$ is always constant with respect to $t$ – it is either 0 or 1. We can substitute then group as follows:

$$\frac{d}{dT}c_n(T) = \left(\frac{1}{T - t_0}\right)\left[f(T)\overline{\phi(1)} - f(t_0)\overline{\phi(0)}\frac{\delta}{\delta T}(t_0)\right] +$$
$$\left(\frac{1}{T - t_0}\right)\left[-\frac{\delta(t_0)}{\delta T}\overline{P} - \left(1 - \frac{\delta(t_0)}{\delta T}\right)\overline{P_t} - \left(1 - \frac{\delta(t_0)}{\delta T}\right)\right]\underbrace{\int_{t_0}^{T}f(t)\left(\frac{1}{T - t_0}\right)\overline{\phi\left(\frac{t - t_0}{T - t_0}\right)}dt}_{c(T)} \tag{A.9}$$

Noting that the final term in this equation contains Eq. A.2, we can simplify further:

$$\frac{d}{dT}c_n(T) = \left(\frac{1}{T - t_0}\right)\left[f(T)\overline{\phi(1)} - f(t_0)\overline{\phi(0)}\frac{\delta(t_0)}{\delta T}\right] +$$
$$\left(\frac{1}{T - t_0}\right)c(T)\left[\frac{-\delta(t_0)}{\delta T}\overline{P} - \left(1 - \frac{\delta(t_0)}{\delta T}\right)\overline{P_t} - \left(1 - \frac{\delta(t_0)}{\delta T}\right)\right] \tag{A.10}$$

Unfortunately, we still have a term $f(t_0)$ that we don't have access to; this is the value of the function at the start of our window. But we have not stored this value; that would defeat the point of an online update in the first place. Instead, we will *approximate it* based on our current coefficient vector and our known basis.

$$c = \langle\phi, f\rangle$$
$$f = \langle\widetilde{\phi}, c\rangle$$
$$f(t_0) = \langle\widetilde{\phi}(0), c(T)\rangle$$

We now have an update rule for $c$ that depends only on the frame $\phi$, the current value of $c(T)$, and the new information from the signal, $f(T)$:

$$\frac{d}{dT}c(T) = \left(\frac{1}{T - t_0}\right)\left[f(T)\overline{\phi(1)} - \widetilde{\phi}(0)c(T)\overline{\phi(0)}\frac{\delta(t_0)}{\delta T}\right]$$
$$-\left(\frac{1}{T - t_0}\right)\left[c(T)\left[\frac{\delta(t_0)}{\delta T}\overline{P} + \left(1 - \frac{\delta(t_0)}{\delta T}\right)\overline{P_t} + \left(1 - \frac{\delta(t_0)}{\delta T}\right)\right]\right] \tag{A.11}$$

### A.1.1 The scaled case

In the case of scaling, $t_0 = 0$ and $\frac{\delta}{\delta T}(t_0) = 0$.

$$\frac{d}{dT}c_n(T) = \left(\frac{1}{T}\right)\left[f(T)\overline{\phi(1)} - \widetilde{\phi}(0)c(T)\overline{\phi(0)}\cancel{\frac{\delta(t_0)}{\delta T}}^{0}\right] \tag{A.12}$$

$$-\left(\frac{1}{T}\right)c(T)\left[\cancel{\frac{\delta(t_0)}{\delta T}}^{0}\overline{P} + \left(1 - \cancel{\frac{\delta(t_0)}{\delta T}}^{0}\right)\overline{P_t} + \left(1 - \cancel{\frac{\delta(t_0)}{\delta T}}^{0}\right)\right] \tag{A.13}$$

$$\frac{d}{dT}c_n(T) = \left(\frac{1}{T}\right)f(T)\overline{\phi(1)} - \left(\frac{1}{T}\right)c(T)(\overline{P_t} + 1) \tag{A.14}$$

The A matrix acts on the coefficient vector c, and B acts on the current input, f(T). Expressed in matrix notation:

$$\frac{d}{dT}c_n(T) = -\frac{1}{T}\underbrace{(\overline{P_t}+I)}_{A}c(T) + \frac{1}{T}\underbrace{\overline{\phi(1)}}_{B}f(T) \tag{A.15}$$

Equivalently,

$$\frac{d}{dT}c_n(T) = -\frac{1}{T}\underbrace{\left(\left\langle\widetilde{\phi}\left(\frac{t}{T}\right), t\phi\left(\frac{t}{T}\right)'\right\rangle + I\right)}_{A}c(T) + \frac{1}{T}\underbrace{\overline{\phi(1)}}_{B}f(T) \tag{A.16}$$

### A.1.2 The translated case

Now $T - t_0 = \theta$ where $\theta$ is the window size, and $\frac{\delta}{\delta T}(t_0) = 1$. Following the same procedure as the previous section:

$$\frac{d}{dT}c_n(T) = \left(\frac{1}{\theta}\right)f(T)\overline{\phi(1)} - \left(\frac{1}{\theta}\right)c(T)\left[\widetilde{\phi}(0)\overline{\phi(0)} + \overline{P}\right] \tag{A.17}$$

$$\frac{d}{dT}c_n(T) = -\frac{1}{\theta}\underbrace{(\overline{P} + \widetilde{\phi}(0)\overline{\phi(0)})}_{A}c(T) + \frac{1}{\theta}\underbrace{\overline{\phi(1)}}_{B}f(T) \tag{A.18}$$

$$\frac{d}{dT}c_n(T) = -\frac{1}{\theta}\underbrace{\left(\left\langle\widetilde{\phi}\left(\frac{t}{\theta}\right), \phi'\left(\frac{t}{\theta}\right)\right\rangle + \widetilde{\phi}(0)\overline{\phi(0)}\right)}_{A}c(T) + \frac{1}{\theta}\underbrace{\overline{\phi(1)}}_{B}f(T) \tag{A.19}$$

## A.2 Experiments

### A.2.1 Datasets

In this paper, we conducted our experiments on these datasets:

**M4 forecasting competition:** The M4 forecasting competition dataset [36] consists of 100,000 univariate time series from six domains: demographic, finance, industry, macro, micro, and other. The data covers various frequencies (hourly, daily, weekly, monthly, quarterly, yearly) and originates from sources like censuses, financial markets, industrial reports, and economic surveys. It is designed to benchmark forecasting models across diverse real-world applications, accommodating different horizons and data lengths. We test on 3,000 random instances.

**Speech commands:** The speech commands dataset [37] is a set of 400 audio files, each containing a single spoken English word or background noise with about one second duration. These words are from a small set of commands, and are spoken by a variety of different speakers. This data set is designed to help train simple machine learning models.

**Wavelet benchmark collection:** Donoho [38] introduced a collection of popular wavelet benchmark signals, each designed to capture different types of singularities. This benchmark includes well-known signals such as Bumps, Blocks, Spikes, and Piecewise Polynomial. Following this model, we synthesize random signals belonging to the classes of bumps, blocks, spikes, and piecewise polynomials. Details and examples of these signals can be found in Appendix A.2.2.

### A.2.2 Wavelet Benchmark Collection

Donoho [38] introduced a collection of popular wavelet benchmark signals, each designed to capture different types of singularities. This benchmark includes well-known signals such as Bumps, Blocks, Spikes, and Piecewise Polynomial.

Following this model, we synthesize random signals belonging to the classes of bumps, blocks, spikes, and piecewise polynomials in our experiments to compare the fidelity of DaubS to legS and fouS, and also to compare the fidelity of DaubT to LegT and FouT.

Figure 9 demonstrates a random instance from each of of the classes of the signals that we have in our wavelet benchmark collection.

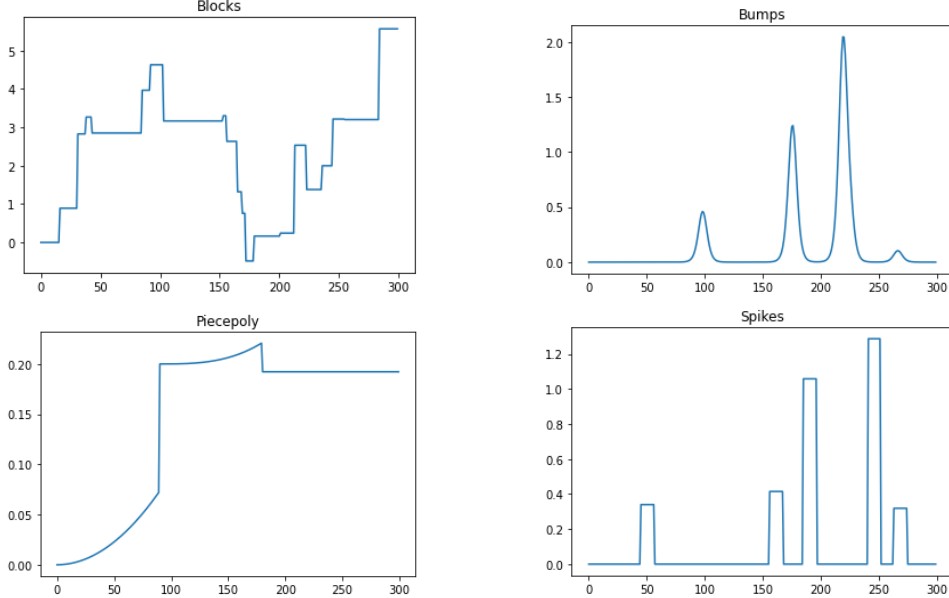

Figure 9: Instances of different signal types in the wavelet benchmark collection. **Top Left:** Blocks is a piecewise constant signal with random-hight sharp jumps placed randomly. **Top Right:** Bumps is a collection of random pulses where each pulse contains a cusp. **Bottom Left:** Piecepoly is a piecewise polynomial signal with discontinuity in the transition between different polynomial parts. **Bottom Right:** Spikes is a collection of rectangular pulses placed randomly with random positive hieght.

### A.2.3 Description of metrics for 'Spikes' and 'Bumps' experiments

- **Peaks Missed** The number of true peaks in the signal is $N_{tp}$, and the number of detected peaks (that is, where the estimated signal surpasses an amplitude threshold $Th_{amp}$), is $N_{dp}$. $N_{dp|tp}$ is the number of detected peaks where a true peak is also within a displacement threshold ($Th_{dis}$) of the detected peak.

$$\text{Peaks Missed} = \left(1 - \frac{N_{dp|tp}}{N_{tp}}\right) \times 100\%$$

- **False Peaks** The metric False Peaks is calculated as the percentage of detected peaks that occurred when there was not a true peak within the displacement threshold. The number of detected peaks when there was no true peak is represented by $N_{dp|\overline{tp}}$.

$$\text{False Peaks} = \frac{N_{dp|\overline{tp}}}{N_{dp}} \times 100\%$$

- **Instance-wise Wins** In each of $K$ time-series instances $S$, Each SSM m gets the instance win over other SSM models if it captures more true peaks than the other models.

$$\text{Instance-wise Wins} = \frac{1}{K} \sum_{k=1}^{K} w_k \times 100\%$$

$$w_k = \begin{cases} 1, & \text{if} \quad \text{Peaks Missed}_m \leq \text{Peaks Missed}_{\text{others}}, \\ 0, & \text{Ow}. \end{cases}$$

In cases where multiple models achieve the same maximum, each tied model receives the credit for that time series instance. As a result, the sum of instance-wise wins for different SSMs may exceed 1.00.

- **Relative Amplitude Error** The relative amplitude error is calculated as the average percent error in the estimated amplitude of detected peaks, including false peaks.

$$\text{Relative Amplitude Error} = \frac{1}{N_{dp}} \left( \sum_{n=1}^{N_{dp|tp}} \frac{|A_{tp,n} - A_{dp|tp,n}|}{A_{tp,n}} \right) \times 100\%$$

- **Average Displacement** The location of a detected peak where a true peak was within a displacement threshold is given by $X_{dp|tp}$. The location of the true peak is denoted as $X_{tp}$.

$$\text{Average Displacement} = \frac{1}{N_{dp}} \sum_{n=1}^{N_{dp}} |X_{tp,n} - X_{dp|tp,n}|$$

### A.2.4 Wavelet frames used for each experiment

Unlike HiPPO-based SSMs, which are fully characterized by their state size $N$, WaLRUS employs redundant wavelet frames that require additional parameters for identification. Once the wavelet frame is defined, the SaFARi framework constructs the unique $A, B$ matrices corresponding to that frame. The key parameters for specifying a redundant wavelet frame in WaLRUS are as follows:

- **Wavelet Function:** Wavelet frames are built from a mother wavelet and a father wavelet, which capture high-frequency details and low-frequency approximations, respectively. Different families such as Daubechies, Morlet, Symlet, and Coifflet provide varied wavelet functions. For this work, we use the D22 wavelet from the Daubechies family.

- **L (Frame Length):** This represents the length of the wavelet frame. Increasing $L$ increases numerical accuracy in the calculation of the $A$ and $B$ matrices at the cost of additional computation time. However, this initial computation need only be performed once, so it is best to choose a large $L$. For the experiments in this work, we set $L = 2^{19}$.

- *Scale min* **and** $N_{\text{eff}}$**:** The minimum scale sets the smallest feature of the signal that can be represented by the frame. This parameter should be chosen based on knowledge about the signal of interest and its component frequencies. Note that the size of the smallest feature is relative to the length of the signal under consideration, so this value may differ under scaling and translating measures.

  For wavelets, *scale min* also controls the effective rank, $N_{\text{eff}}$. Each new lower scale introduces a factor of two in the effective rank of the frame, owing to the additional shifted elements in each scale. Fig. 3 shows two scales, where there are 3 father wavelets ($\phi_0$) and 3 coarse-scale mother wavelets ($\psi_1$). The next scale introduces 6 scaled and shifted mother wavelets ($\psi_2$), the next would include 12, and so on. Table 3 also illustrates this pattern, with *scale min* of 0 corresponding to $N_{\text{eff}}$ of $2^6$, *scale min* of $-1$ corresponding to $N_{\text{eff}}$ of $2^7$, and so on, with some margin of error for numerical accuracy and truncation.

  Our code includes another variable, *scale max*. Since smaller scales can also combine to represent larger scales, *scale max* in fact has no impact on $N_{\text{eff}}$ (see [29] for further information). Fig. 10 demonstrates on an example implementation that varying scale max does not impact the size of $N_{\text{eff}}$. It is also easily shown that varying scale max results in the same diagonalized A; see our code supplement. Adding coarser scales can help improve numerical accuracy in the calculation of A, however. We do not include scale max in Table 3, but we do provide it in our code with each experiment for reproducibility.

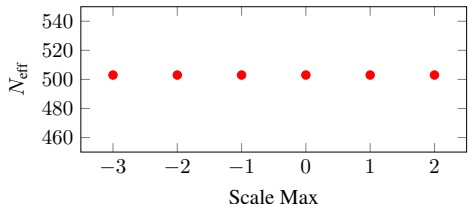

Figure 10: Effective Rank of WaLRUS $A$ matrix with Scale Min=-3, shift=0.01

- **Shift:** At scale $i$, $2^{-i}m$ overlapping shifts are applied to the wavelets, where $0 < m \le 1$ is a shift constant. Setting $m = 1$ corresponds to dyadic shifts. As our wavelet frames typically

| Experiment | Basis/Measure | scale min | shift | $N_{\text{eff}}$ |
|---|---|---|---|---|
| Scaled M4 | WaveS | -3 | 0.01 | 501 |
|  | LegS | - | - | 500 |
|  | FouS | - | - | 500 |
| Scaled Speech | WaveS | -5 | 0.01 | 1995 |
|  | LegS | - | - | 1995 |
|  | FouS | - | - | 1995 |
| Scaled synthetic | WaveS | -3 | 0.01 | 501 |
|  | LegS | - | - | 500 |
|  | FouS | - | - | 500 |
| Scaled peak detection | WaveS | 0 | 0.01 | 65 |
|  | LegS | - | - | 65 |
|  | FouS | - | - | 65 |
| Translated M4 | WaveT | -1 | 0.01 | 128 |
|  | LegT | - | - | 128 |
|  | FouT | - | - | 128 |
| Translated Speech | WaveT | -3 | 0.0025 | 500 |
|  | LegT | - | - | 500 |
|  | FouT | - | - | 500 |
| Translated synthetic | WaveT | -1 | 0.01 | 128 |
|  | LegT | - | - | 128 |
|  | FouT | - | - | 128 |
| Translated peak detection | WaveT | 0 | 0.01 | 65 |
|  | LegT | - | - | 65 |
|  | FouT | - | - | 65 |

Table 3: Parameters for the redundant wavelet frame used by WaLRUS in different experiments. All of the above experiment share the parameters $L = 2^{19}$, and $\text{rcond} = 0.01$.

only contain a few dilation levels, using $m = 1$ can mean that the constructed set of vectors no longer satisfies the frame condition, and is lossy. We choose a small value (0.01 for most experiments), and tune this as needed.

- **rcond:** This parameter controls the numerical stability of the pseudo-inverse calculation for the dual frame. Singular values smaller than $\text{rcond} \times \sigma_{\max}$ are discarded during the inversion process to maintain numerical stability.

Note that all the above parameters are solely to identify the redundant wavelet frame, and that WaLRUS does not introduce any new parameters. Table 3 summarizes the settings for all experiments, alongside the SSM sizes for HiPPO-Legendre and HiPPO-Fourier.

### A.2.5  Computational resources

Within the scope of this paper, no networks were trained and no parameters were learned. Only CPU resources were utilized, but speed could be improved with parallel resources on a GPU. Using WaLRUS to find representation has two different stages:

- Pre-computing: Computing SSM $A$ matrices and diagonalizing them. This step can be computationally intensive, but need only be calculated once.
- Computation: Using SSM $A$ matrices to find representations of signals.

For all our experiments except Scaled-Speech, the pre-computing stage takes less than 10 minutes. For scaled-speech, the pre-compute time is on the order of hours. Once the A matrices are computed and stored, run time is the same for all experiments.

