# OpenReview forum: "WaLRUS: Wavelets for Long range Representation Using State Space Methods"
_NeurIPS.cc/2025/Conference — NeurIPS 2025 poster_

### Official Review · Reviewer_tu5D · 2025-07-01

**Clarity:** 2
**Significance:** 2
**Originality:** 2
**Rating:** 4
**Confidence:** 4

**Summary:**

The paper proposes and demonstrates replacing Legendre and Fourier basis sets for State Space Modeling of dynamic trajectories with Wavelet basis functions. The theory is described in general terms and experiments are performed on benchmark problems involving many dynamic systems and spike recognition problems. Comparisons with other basis set formulations (Legendre polynomials and Fourier series) for State Space Modeling show that the wavelet basis often outperforms.

**Questions:**

Can you provide a clearer overview of the theory, to include precise justifications of all claims?

Can you provide comparisons with other, non state-space models?

Can you provide a code supplement for reproducibility?

**Ethical Concerns:**

["NO or VERY MINOR ethics concerns only"]

**Final Justification:**

I missed the repo in the footnote. I am also not a part of the SSM culture -- maybe that community is large enough that an advance like this completely within that context is enough for acceptance at NeurIPS. I would assume that some comparisons with methods outside of SSM would be requried but I'll leave that to the chairs.

**Limitations:**

yes

**Quality:**

2

**Strengths And Weaknesses:**

The wavelets clearly outperform the comparison bases on a broad number of test problems.

However, there is not much theoretical development. In particular it is not shown how to calculate the transition matrices $A$ and $B$ using the wavelet basis. It is claimed that wavelet bases always yield diagonalizable state-space representations, but this is not justified. Furthermore the introduction of the state space method leaves something to be desired, as it is unclear how the minimization on line 70 yields a state vector.

More serious is the fact that only state-space methods are compared with each other, while competing methods (for instance, RNNs, but also many other methods for dynamic systems, in particular Markov methods) are left out. Finally there seems to be no code supplement for reproducibility purposes.

---

> ### Author Rebuttal · Authors · 2025-07-31
>
> Dear Reviewer,
>
> Thank you for your feedback.  We would like to address some of your concerns about this work in order to strengthen and improve our paper.  Brief summaries of your notes are in bold, with our responses below.
>
> **“There seems to be no code supplement for reproducibility purposes.”**
>
> A link to the code supplement was provided in our paper twice.  Please see the footnote on page 1, and the NeurIPS checklist, item 5.
>
> **“...it is not shown how to calculate the transition matrices A and B using the wavelet basis.”**
>
> Equations 3 and 4 give the construction of A and B explicitly.  These equations and their derivations can be found in reference [17], as noted in the text.  There is also an algorithmic implementation in our code.
>
> The construction of wavelet frames is well-covered in the literature (see references [18],[20],[29],[30]).  Figure 2 contains an illustration of a partial Daubechies wavelet frame, providing readers with key information (scales, shifts, and function shape) required to understand the contribution of this paper, without needing full knowledge of the vast literature on wavelet frame construction.
>
> To facilitate easy access to key information for our readers, we have also added a section in the Appendix of our revised manuscript with details from [17] on the derivations of A and B.
>
> **“It is claimed that wavelet bases always yield diagonalizable state-space representations, but this is not justified”**
>
> We did not claim that wavelet bases “always yield” diagonalizable SSMs; we were careful to note that wavelet-based SSMs were found to be *empirically* stably diagonalizable (line 47-51) and *observed* to be diagonalizable in our experiments (line 137). Prior work has not proved any theory regarding which bases/frames or measures produce stably diagonalizable $A$ matrices, and have thus far only reported empirical observations, as we have done.
>
> Critically, the only significant previous work on this subject (HiPPO) found that even at fairly low rank, Legendre-based matrices are *not* stably diagonalizable.  This problem was a major roadblock in the development of S4 and ultimately Mamba, and provided the motivation for multiple subsequent follow-up papers on how to make diagonal *approximations* to the Legendre-based $A$ matrix (see refs [12-14]).  In fact, if one examines the code for the state-of-the-art Mamba model (https://github.com/state-spaces/mamba), the SSM layers are initialized with one of these approximations, and not the true $A$ matrix.  Thus, we believe it is valuable to note our empirical findings of diagonalizability, even in the absence of a full theoretical proof.
>
> To make very clear the limits of our claims, we have made the following wording adjustments in our revised manuscript:
>
> Line 47-51, Section 1: “Empirical results demonstrate … WaLRUS enjoys diagonalizability, which is the key enabler of efficient convolution-based implementations and parallel computation [13, 14].”
>
> Change to → “Empirical results demonstrate … WaLRUS has been consistently experimentally observed to be stably diagonalizable, which is the key enabler of efficient convolution-based implementations and parallel computation [13, 14].”
>
> Line 137, Section 3.2: “We observe that both scaled and translated WaLRUS are stably diagonalizable.”
>
> Change to → “We observed that each of the scaled and translated WaLRUS SSMs we implemented, regardless of dimension, were stably diagonalizable.  Further research is required to determine whether Daubechies wavelets will always yield diagonalizable SSMs.”
>
> While a complete theoretical proof of diagonalizability is outside the scope of this particular work, we agree that it would be a significant contribution to the literature on this topic.  We hope that our paper inspires members of our community to join us in exploring this problem.
>
> **“Only state-space methods are compared with each other, while competing methods (e.g., RNNs, Markov methods) are left out.”**
>
> RNNs and Markov methods are not competing methods with online basis projection.  Our paper proposes an alternative basis for state-of-the-art *function approximation* SSMs that play a key role in some of the fastest-growing models in ML today; it does not propose a *time-series predictor*.
>
> HiPPO / SaFARi / WaLRUS only store *previous* information by iteratively (and deterministically) projecting a signal onto a basis; probabilistic methods such Markov methods are designed for predictive tasks.  Most importantly, neither RNNs nor Markov methods produce a model compatible with use in S4, Mamba, and Mamba’s variants, a driving motivation for this work.  Thus, the only baseline for comparison is existing SSMs for function approximation (namely HiPPO).
>
> **“it is unclear how the minimization on line 70 yields a state vector”**
>
> The results in line 70 were a key finding of the original HiPPO paper [9]; we do not reproduce it in our paper, only noting the results pertinent to our work.  To ensure that this paragraph does not cause confusion for readers unfamiliar with HiPPO, we have made the following wording change in our revised manuscript, removing the reference to the minimization:
>
> Line 70: “To resolve these issues, the HiPPO framework [9] casts online function approximation as a continuous projection of the input u(t) onto a linear combination of the given basis functions G. At every time T, it ~~solves ming (T )∈G ∥uT − g (T) 70 (t)∥µ , producing~~ produces a compressed state vector c(T) that satisfies the update rule:”
>
> We thank the reviewer again for their time and the opportunity to clarify and improve our paper.

---

> > ### Comment · Reviewer_tu5D · 2025-08-05
> > **a few responses**
> >
> > First of all sorry for missing the repo in the footnote.
> >
> > I am not that familiar with state space models, but am pretty unsatisfied with the explanation that these methods are not about prediction. State-Space was developed for control which is absolutely about prediction. If online updates (function approximation) is what supposedly makes SSM distinct, I'll point out that there are plenty of other methods that include online function estimation, such as Koopman operator methods (see https://www.nature.com/articles/s41598-023-49045-w and refs therein).
> >
> > I am increasing my rating for the missed repo but I think I have to be a dissenting voice here for the tunnel-vision focus on SSMs. Generally I am all for finding a better basis and popping it into an existing framework but here we have a situation in which neither the basis nor the framework are novel, only the combo -- as another reviewer noted.

---

> > > ### Author Response · Authors · 2025-08-07
> > > **Addressing your comments on scope, novelty, and SSM usage**
> > >
> > > Dear Reviewer,
> > >
> > > We thank you for the additional feedback.  We presume you are also satisfied with our explanations regarding your initial questions on the construction of $A$ and $B$ and the state vector, and the relevance of empirical results on diagonalizability.  However, there still seems to be some confusion on the purpose and formulation of SSMs for predictive versus memorization tasks, and some more general critiques on the scope and relevance of this work.  We have responded to your comments below in two sections: the first addressing high-level concerns, and the second on the specifics of SSMs.
> > >
> > > **1. High-level concerns on scope and novelty**
> > >
> > > **“I think I have to be a dissenting voice here for the tunnel-vision focus on SSMs”**
> > >
> > > We disagree that focusing our paper on SSMs constitutes “tunnel vision”; these SSM structures are already at the heart of state-of-the-art machine learning models, hence why we are interested in studying them.  Ours is only the 3rd paper (to our knowledge) to address their construction, and we compare our work to the only extant relevant baselines.
> > >
> > > **“we have a situation in which neither the basis nor the framework are novel, only the combo”**
> > >
> > > One could say the same of JPEG 2000!  Combining established components often yields new capabilities, theoretical clarity, or state-of-the-art results. (Even NeurIPS guidelines explicitly recognize “a novel combination of existing techniques” as original.)  Moreover, we do not believe that the introduction of a novel framework or novel basis is useful or reasonable for this work.  The SSM framework is studied specifically for its relevance in state-of-the-art ML.  The invention of an entirely new basis of orthogonal functions would be a significant contribution to mathematics, and not appropriate for the scope of a conference paper.
> > >
> > > The combination of an SSM memory unit over Daubechies wavelets *is* novel and relevant: our work is the very first paper to offer an explicit alternative to the underlying machinery of HiPPO-based models.  This implementation is nontrivial, as evidenced by the fact that over 1,000 Mamba-based papers since 2023 (mostly on arXiv) have never even attempted it.  Even papers such as those referenced by Reviewer UUkH, which purport to leverage an alternate basis (wavelets), are unable to do so at the architectural level; they resorted to applying a decomposition to the input only.
> > >
> > > **2. Specifics of SSMs, and the distinction between prediction and memory tasks**
> > >
> > > **“State-Space was developed for control which is absolutely about prediction”**
> > >
> > > State-space models are just a mathematical representation of dynamical systems, like ODEs.  While they were introduced in the 1950s initially for control problems via the Kalman filter, the subsequent 75 years have seen them applied to myriad tasks, including function approximation, filtering, smoothing, state estimation, and system identification – not all of which involve prediction, or even control, for that matter.  Most recently, they have been re-purposed within ML models to summarize historical information using GPU-friendly convolution operations, and this is the context in which our work resides (see Introduction and Background, and refs [9-17]).
> > >
> > > Consider a simplified system represented by $\frac{dx}{dt} = Ax(t) + Bu(t)$.  Depending on the problem or application, one might be interested in *controlling* $x(t)$, or *predicting* future values of $u(t)$, or *learning* the state transition matrix $A$, etc.  It would be nonsensical to compare these; for example, the accuracy of an estimation for $A$ is irrelevant for a problem where $A$ is known.
> > >
> > > In our work, following HiPPO and SaFARi, we construct $A$ and $B$ deterministically (Eqns 3-4), and iteratively project the history of a signal $u(t)$ onto a basis with coefficients given by $x(t)$.  (See equations 1-4, and Sec. 2).  Appropriate baselines for comparison are other methods that project the history of $u(t)$ onto a basis.
> > >
> > > **“If online updates (function approximation) is what supposedly makes SSM distinct, I'll point out that there are plenty of other methods that include online function estimation, such as Koopman operator methods”**
> > >
> > > This comparison conflates a system and its input.
> > >
> > > The referenced paper on Koopman operators is focused on online learning of the *operator*, analogous to $A$ in our simplified formulation above.  It assumes that a nonlinear system already exists and we wish to find a linear representation of it.  HiPPO, SaFARi, and WaLRUS, by contrast, design the operator $A$ (Sec. 2, eqs 3-4, and refs [9-17]); the function being estimated is the *input*.  These are fundamentally different problems with fundamentally different solution methods, and it is unclear to us how they would be compared.
> > >
> > > We thank the reviewer again for their time and perspective, and hope that the above clarifications have been helpful.

---

### Official Review · Reviewer_obbg · 2025-07-03

**Clarity:** 3
**Significance:** 2
**Originality:** 2
**Rating:** 5
**Confidence:** 3

**Summary:**

The paper builds upon the established SaFARi framework, which does frame-agnostic representations for function approximation using state-space models. The authors introduce WaLRUS, a novel extension that replaces the Legendre or Fourier frames in SaFARi with wavelet frames. They provide an analysis of the theoretical advantages and potential limitations of this approach, including
a discussion on computational complexity and approximation error. The experimental evaluation, done on both real and synthetic datasets as well as a new "spike detection" task, demonstrates the overall superiority of wavelet frames.

**Questions:**

1. The critique in section 3 on other frames seems to focus on the Legendre translation (LegT) frame, other frames occupy less room. Can you balance this more?
2. The metrics for the "isolated spikes" task are only described with an image. Could you provide some formulas as well?
3. What is the effect of varying the effective dimension N_eff? Could you plot MSE over N_eff?

**Ethical Concerns:**

["NO or VERY MINOR ethics concerns only"]

**Final Justification:**

The rebuttal has improved clarity of the proposed method and my questions were overly answered.
I stay with the finding that the improvements in MSE exist but are not that high. The relevance of those improvements to down stream tasks remain not fully convincing. Nevertheless, replacing the Legendre or Fourier frames in SaFARi with wavelet frames is interesting and the results are thorougly validated. So I stay with my positive assessment.

**Limitations:**

yes

**Paper Formatting Concerns:**

- line 152 typo: "accuraty"
- some white space around figures

**Quality:**

3

**Strengths And Weaknesses:**

Strengths:
- well written
- clear objective of the paper
- theoretical motivated
- empirically proven

Weaknesses:
- The authors switch a part in an existing framework. While this is shown to be better, the overall novelity is not of the highest quality.
- While outperforming existing methods, the shown improvements in MSE are not that high. The relevance of those improvements to down stream tasks ramain uncertain.
- Wavelet frames come with more parameters, e.g. scale, shift N_eff. It is totally unclear how to set them. There are some justifications for the set parameters, but it is still unclear how to set them properly. This is however mentioned in the limitations.
- A.1.3 lists all the chosen parameters, which vary from experiment to experiment. That seems like hyperparameter optimization and making the significance of the result somewhat questionable.

---

> ### Author Rebuttal · Authors · 2025-07-31
>
> Dear Reviewer,
>
> Thank you very much for your thorough and insightful feedback.  We would like to address some of your concerns about this work in order to strengthen our paper.  Brief summaries of your feedback are in bold, with our responses below.
>
> **“The authors switch a part in an existing framework. While this is shown to be better, the overall novelty is not of the highest quality…  the shown improvements in MSE are not that high. The relevance of those improvements to downstream tasks remain uncertain.”**
>
> We have framed our work as a drop-in replacement for SSM-based ML models as motivation and context, but this work also stands alone as a lightweight online function approximation tool.  More importantly, it is *the very first work* offering a replacement for the Legendre-based SSM that is taken for granted when utilizing off-the-shelf Mamba/S4 implementations.  To date, no other work has ever gone back to first principles to understand the functionality of the core machinery running these models, let alone fundamentally alter it.
>
> Based on results from prior and current work, we believe that using better-performing function-approximating SSMs in neural networks should improve performance in downstream tasks; however, we first must show how and why these SSMs perform better on their own.  The focus of this paper is thus on developing a core building block that will enable those downstream applications, and hopefully be of use and interest to members of the community.
>
> Our revised manuscript includes a new figure--a simplified block diagram illustrating the relationship between our work, HiPPO, SaFARi, and Mamba.  We have also included some additional text in Section 2 that contextualizes our work in the space of Mamba and other such models:
>
> "Some recent work [R1,R2] has conceptually connected the use of wavelets and SSM-based models (namely Mamba). These efforts are fundamentally distinct from ours in that they perform a multi-resolution analysis on the *input to the model only*. No change is made to the standard Mamba SSM layer; it is still using Legendre-polynomial based SSMs.
>
> This work, on the other hand, is the first to challenge the ubiquity of the Legendre-based SSM, and present alternative wavelet-based machinery for the core of powerful models like Mamba.  WaLRUS could be used as a drop-in replacement for any existing SSM-based framework.  However, before simply substituting a part in a larger system, we must first justify how and why a different SSM can improve performance. This paper presents a tool that stands alone as an online function approximator, and also provides a foundational building block for future integration in SSM-based models.”
>
> **“Wavelet frames come with more parameters, e.g. scale, shift N_eff. It is totally unclear how to set them. There are some justifications for the set parameters, but it is still unclear how to set them properly. This is however mentioned in the limitations.”**
>
> Scales for Daubechies wavelets are always dyadic, scaling by a factor of 2; we have updated the equation in Section 2.1 to make this clearer:
> $$
> W(j,k) = \int_{-\infty}^\infty f(t) \psi^*_{j,k}(t) \, dt,
> \quad
> \psi_{j,k}(t) = \frac{1}{\sqrt{2^{-j}}} \psi\left(\frac{t-k}{2^{-j}}\right)
> $$
> The choice of shift amount introduces some interesting complexity that we agree could be given deeper treatment, and we have added the following text to Section 3:
>
> “To construct the frame, we use the usual dyadic scaling for multiresolution analysis; that is, scaling the mother wavelets by a factor of two at each level.  For each scale, different shifts along the x-axis are introduced. Compressive wavelet frames are truncated versions of wavelet frames that contain only a few of the coarser scales, and introduce overlapping shifts to keep the expressivity and satisfy the frame condition (See Mallat, [18]).  The interplay between the retained scales and the minimum required overlap to maintain the expressivity is extensively studied in the wavelet literature [18,20,30].  If there is excess overlap in shifts, the wavelet frame becomes redundant, and redundancy has advantages in expressivity and robustness to noise.
>
> The exact relationship between the amount of overlap and the resulting dimensionality of the A matrix depends not only on the overlap of the shifts, but also on the type (and order) of chosen wavelet, and number of scales.  Determining the “optimal” overlap or N_eff is application-specific, and an area for future research.”
>
> **“A.1.3 lists all the chosen parameters, which vary from experiment to experiment. That seems like hyperparameter optimization and making the significance of the result somewhat questionable.”**
>
> Some parameter choices are heuristic and some are codependent, but none were chosen to be optimized, or to otherwise unfairly advantage one method over another.  We acknowledge that the varying parameters as presented in Table 3 could reasonably raise this concern for readers, however, and so we have added additional text to clearly explain how these parameters were chosen.  We have also simplified our table to remove the wavelet type (D22, which is the same for every experiment in this work and is already described in the same appendix), and the “scale max” variable, which does not impact the rank of the frame, as explained in the added text below.
>
> “*Scale Min* and $N_{\rm{eff}}$: The minimum scale sets the smallest feature of the signal that can be represented by the frame. This parameter should be chosen based on knowledge about the signal of interest and its component frequencies, as one would for a filter bank.  Note that the size of the smallest feature is relative to the length of the signal under consideration, so this value may differ under scaling and translating measures.
>
> *Scale min* also controls the effective rank, $N_{\rm{eff}}$. Each additional lower scale introduces a factor of two in the effective rank of the frame, owing to the additional shifted elements in each scale.  To illustrate, Fig. 2 shows two scales, where there are three father wavelets ($\phi_0$) and three coarse-scale mother wavelets ($\psi_1$). The next scale introduces six scaled and shifted mother wavelets ($\psi_2$), the next would include 12, and so on. Table 3 also shows this pattern, with a minimum scale of 0 corresponding to $N_{\rm{eff}}$ of $2^6$, minimum scale of $−1$ corresponding to $N_{\rm{eff}}$ of $2^7$, and so on, with variance resulting from the shift combined with truncation of frame elements during construction.
>
> Our code includes another variable, *scale max*. Since smaller scales can also combine to represent larger scales, *scale max* in fact has no impact on $N_{\rm{eff}}$.  (See Mallat [18] for further details on this topic.)  Adding coarser scales can help improve numerical accuracy in the calculation of $A$, however.  We do not include *scale max* in Table 3, but we provide it in our code with each experiment for reproducibility."
>
> To support this additional text, we also have generated some quantitative results.  Fig. 10 (in the appendix of our revised paper) takes an example implementation of WaLRUS and varies *scale max*, and we observe that the size of $N_{\rm{eff}}$ remains constant.  It is also easily shown that varying scale max results in the same diagonalized $A$ matrix.
>
> **“What is the effect of varying the effective dimension N_eff? Could you plot MSE over N_eff?”**
>
> We agree that analyzing the impact of varying $N_{\rm{eff}}$​ on MSE would be insightful.  However, due to limited time, we cannot include this experiment in the current revision, and plan to pursue it in future work.
>
> **“The critique in section 3 on other frames seems to focus on the Legendre translation (LegT) frame, other frames occupy less room. Can you balance this more?”**
>
> We have added the following text in Section 3 of our revised manuscript after the discussion of the translated Legendre kernel:
>
> "The translated Fourier (FouT) kernel primarily suffers from the opposite problem: substantial nonzero elements outside the kernel window indicate that LegT struggles to effectively "forget'' historical input values. Thus contributions from input signals outside the sliding window appear as representation errors.  LegT also has this problem, to a lesser extent--see area A of Fig. 3."
>
> We keep the primary focus on Legendre-based kernels, as this is the most ubiquitous (even if hidden!) SSM used today in ML architectures.
>
> **“The metrics for the "isolated spikes" task are only described with an image. Could you provide some formulas as well?”**
>
> Yes; we have added a section in the appendix of our revised manuscript with these formulas.
>
> We thank the reviewer again for their thoughtful questions, which spurred interesting discussions among the authors and provided concrete ways to better communicate the content and context of this work.  We believe our updated manuscript is significantly improved, and look forward to any further feedback you would provide.
>
> ---
>
> References:
>
> [R1] Zou, W., Gao, H., Yang, W., & Liu, T. (2024, October). Wave-mamba: Wavelet state space model for ultra-high-definition low-light image enhancement. In Proceedings of the 32nd ACM International Conference on Multimedia (pp. 1534-1543).
>
> [R2] Zhang, T., Zhu, Y., Zhao, J., Cui, G., & Zheng, Y. (2025). Exploring state space model in wavelet domain: An infrared and visible image fusion network via wavelet transform and state space model. arXiv preprint arXiv:2503.18378.

---

### Official Review · Reviewer_UUkH · 2025-07-06

**Clarity:** 2
**Significance:** 3
**Originality:** 3
**Rating:** 4
**Confidence:** 3

**Summary:**

The paper proposes WaLRUS, a wavelet-based formulation of SSMs. For this the authors start from the SaFARi framework and leverage Daubechies wavelet frames. Experimental evaluations of WaLRUS on online function approx tasks show improved performance compared to two baselines: HiPPO-LegT and HiPPO-FouT.
The paper argues that the wavelet-based kernel has better localization, and is better at resolving sparse signal (e.g two close by signal peaks as shown in Fig. 1).

Disclaimer: I'm not an expert either in SSMs nor in wavelets. Thus, I may not be familiar with some shared assumptions in these subfields. However, my review may still add value as a sample of the potential attendant distribution (so for exposure, understandability etc).

**Questions:**

1) Why does the paper omit a direct comparison with SaFARi, even though WaLRUS is built on it? Could the reported gains simply stem from inheriting SaFARi’s advantages rather than from the wavelet basis itself?

2) Do the current benchmarks really demonstrate WaLRUS’s benefits over existing SSMs (and maybe even Transformers)? I believe an experiment on more community-relevant settings a small-scale long-context language-model or long biological sequences could provide a clearer picture.

3) Given earlier papers that combine wavelets with SSMs (see [1, 2] above), what specific gap does WaLRUS fill, and how does its contribution differ from or improve upon these prior approaches? Would these need to be shown experimentally as well?

4) How reusable is the proposed Walrus code? I.e. from reading the paper I'm not sure how easy it will be to try this approach out as drop-in replacement in existing SSM frameworks.

**Ethical Concerns:**

["NO or VERY MINOR ethics concerns only"]

**Final Justification:**

Rebuttal adressed my concerns. Willing to raise my score by 1; but still not a super strong submission in my view due to limited clarity.

**Limitations:**

yes

**Quality:**

2

**Strengths And Weaknesses:**

Strengths:
- Walrus seems to be quite efficient: under certain conditions the runtime over a sequence of length L is claimed to be O(L) using parallel processing.
- Overall, I think it's a great idea to propagate insights from signal processing / numerical analysis like wavelets through to recent ML techniques like SSM.
- To my understanding, the wavelet representations add the nice property of compact support - whereas many other bases would lead to non-zero values also outside of the window of interest.

Weaknesses:
On a high level, I'm confused about three main parts:
1) The narrative of the abstract/intro is: There is HiPPO --> SaFARi --> WaLRUS (based on SaFARI). But the experiments compare WaLRUS only against two HIPPO variants. Why not compare against the direct baseline SaFARi? (as any benefit could be due to the advantage of building on SaFARi otherwise.
2) The experimental evaluations are on tasks that I personally do not find super relevant to showcase the benefit of a new SSM variant:
- i.e. evaluating on some basic time series dataset and peak detection may be fair - but I expect the community would be more curious to see if walrus can offer benefits in the typical SSM vs transformer settings (e.g. long-range / long-context language modelling, long biological sequences etc.). Why not do a tiny-scale language model experiment by training small models (e.g. GPT-2 scale) regular transformer vs SSM (best prior) vs SSM (Walrus).
3) There seem to be few works [1], [2], that do Wavelets with SSM, I haven't checked them in careful detail, but it begs the quesiton whether Wavelet + SSM is novel - or what are the gaps that this works fills exactly?

Smaller points:
- Fig 1: only shows HiPPO (two version) against WalRus - but not SaFARi baselines.
- It would be nice to provide a bit more exposure/explanation to Eq 3,4 which use notation that not everbody in this ML venue may be familiar with.
- Purely aesthetic: but I would remove the vertical lines from Tab 2

References:
[1] https://arxiv.org/pdf/2408.01276
[2] https://arxiv.org/pdf/2503.18378

---

> ### Author Rebuttal · Authors · 2025-07-31
>
> Dear Reviewer,
>
> Thank you for your thorough and insightful feedback.  We would like to address some of your concerns about this work in order to strengthen our paper and better tailor it to a NeurIPS audience.  Brief summaries of your feedback are in bold, with our responses below.
>
> **“It would be nice to provide a bit more exposure / explanation to Eq 3,4 which use notation that not everybody in this ML venue may be familiar with.”**
>
> We agree that since this is very recent work that readers may not be familiar with, it will be helpful to provide deeper insight into these equations.  We have written a new appendix section which provides a quick and convenient reference for readers on the derivations of Eqns 3 and 4 following reference [17].  This section starts from a simple projection of a signal onto a basis via inner product, and develops the full SSM models step by step with text narrative to build intuition along with the mathematical framework.
>
> **“The narrative of the abstract/intro is: There is HiPPO --> SaFARi --> WaLRUS (based on SaFARI). But the experiments compare WaLRUS only against two HIPPO variants. Why not compare against the direct baseline SaFARi?”**
>
> We have added clarifying text to Section 1 to emphasize the point that SaFARi is a method to generate models, whereas WaLRUS is a model instance:
>
> “In this paper, we leverage the SaFARi method with wavelet frames to introduce a new SSM instance, WaLRUS (Wavelets for Long-range Representation Using SSMs).”
>
> We have also created a block diagram to illustrate the relationships between WaLRUS, HiPPO, SaFARi, and Mamba, which also helps to highlight the unique contributions of this paper and the distinction from other works.
>
> **“How reusable is the proposed Walrus code? I.e. from reading the paper I'm not sure how easy it will be to try this approach out as drop-in replacement in existing SSM frameworks.”**
>
> WaLRUS is indeed a drop-in replacement for existing SSM frameworks using HiPPO; the new figure in our revision explains this.  Our code generates A and B matrices of size NxN and Nx1 respectively, which can be directly substituted into any code that uses SSM layers made from A and B matrices (code supplement provided in footnote on p.1).
>
> **“There seem to be few works [1], [2], that do Wavelets with SSM… it begs the question whether Wavelet + SSM is novel - or what are the gaps that this works fills exactly?”**
>
> We did not cite extensively from the vast and growing literature on Mamba, since it is outside the scope of our paper, except as motivation and contextualization.  Yet, given the overlap in the titles of these papers and the focus of our work, we expect that other readers will have the same question.  Therefore, we have include these citations in Section 2 with the following text:
>
> "Some recent work [32, 33] has conceptually connected the use of wavelets and SSM-based models (namely Mamba). These efforts are fundamentally distinct from ours in that they perform a multi-resolution analysis on the input to the model only. No change is made to the standard Mamba SSM layer; it is still using Legendre-polynomial based SSMs.
>
> This work, on the other hand, is the first to challenge the ubiquity of the Legendre-based SSM, and present alternative wavelet-based machinery for the core of powerful models like Mamba.  WaLRUS could be used as a drop-in replacement for any existing SSM-based framework.  However, before simply substituting a part in a larger system, we must first justify how and why a different SSM can improve performance. This paper presents a tool that stands alone as an online function approximator, and also provides a foundational building block for future integration in SSM-based models.”
>
> **“Do the current benchmarks really demonstrate WaLRUS’s benefits over existing SSMs (and maybe even Transformers)? I believe an experiment on more community-relevant settings a small-scale long-context language-model or long biological sequences could provide a clearer picture.” “The experimental evaluations are on tasks that I personally do not find super relevant to showcase the benefit of a new SSM variant.”**
>
> WaLRUS, like HiPPO, does not learn any parameters; it is a purely signal-processing approach that does not compare directly to learned models like transformers.  It holds the promise of building better machinery for ML models like Mamba, but it also stands on its own as a function approximation technique.  Our goal in choosing experimental cases was to highlight the power of WaLRUS in function approximation as a first step in developing a new tool, and as such, the only appropriate comparison is to HiPPO.  We again use our new diagram to emphasize this focus.
>
> The use cases you suggested – language models and biological sequences – are excellent candidates for future work, e.g. by comparing a WaLRUS-based Mamba to a HiPPO-based Mamba.  In fact, we hope that this work will inspire the community to apply WaLRUS in such scenarios by building on this foundational work.
>
> **Formatting of Table 2:**
> We agree that without vertical lines it is much nicer; we have removed these in our most recent revision.
>
> We believe our updated manuscript is significantly improved and better contextualizes the work.  We thank the reviewer again for their time and for the valuable feedback on our paper, and look forward to any additional comments you would provide.

---

> > ### Comment · Reviewer_UUkH · 2025-08-03
> > **Thanks for clarifying.**
> >
> > Thanks for the comments. My main concerns and questions have been answered. I'm willing to raise the score to 4, though I'm quite uncertain about my assessment.

---

### Decision · Program_Chairs · 2025-09-17

**Decision:**

Accept (poster)

**Comment:**

The paper proposes an extension of the SaFARi framework, using state-space models for long-range representation learning. The method offers improvements in representing non-smooth and transient signals compared to HiPPO models.

The reviewers found the approach to be interesting, finding that replacing the Legendre or Fourier frames in SaFARi with wavelet frames does result in a performance improvement. They also appreciated the efficiency of the method. During the discussion phase, questions were raised about the relevance of the experimental setup to validate the claims and the chosen contenders. However, the authors clarified that "WaLRUS, like HiPPO, does not learn any parameters; it is a purely signal-processing approach that does not compare directly to learned models like transformers" and is to be used for function approximation. Thus, they convincingly argued that the comparison against HIPPO is sufficient to validate the claims. The authors also answered questions about the parameters of the model and about wavelet-based SSMs being "empirically stably diagonalizable" diagonalizable state-space representations. Finally, all reviewers determined that the contributions of the paper are over the threshold for acceptance.

As long-term modeling of time series is an important problem and this paper has some degree of methodological novelty and interesting insights, I recommend its acceptance at NeurIPS.